# Effects of the In Ovo and Dietary Supplementation of L-Ascorbic Acid on the Growth Performance, Inflammatory Response, and Eye L-Ascorbic Acid Concentrations in Ross 708 Broiler Chickens [note 1]

**DOI:** 10.3390/ani12192573

**Published:** 2022-09-27

**Authors:** Ayoub Mousstaaid, Seyed Abolghasem Fatemi, Katie Elaine Collins Elliott, April Waguespack Levy, William Wadd Miller, Patrick D. Gerard, Abdulmohsen Hussen Alqhtani, Edgar David Peebles

**Affiliations:** 1Department of Poultry Science, Mississippi State University, Starkville, MS 39762, USA; 2Poultry Research Unit, USDA-ARS, Starkville, MS 39762, USA; 3DSM Nutritional Products, Parsippany, NJ 07054, USA; 4Advanced Animal Eye Care, 3308 Old West Point Road, Starkville, MS 39759, USA; 5Department of Mathematical Sciences, Clemson University, Clemson, SC 29634, USA; 6Department of Animal Production, King Saud University, Riyadh 11451, Saudi Arabia

**Keywords:** in ovo injection, inflammatory response, growth performance, L-ascorbic acid, eye L-AA concentration

## Abstract

**Simple Summary:**

L-ascorbic acid (**L-AA**), known as vitamin C, is involved in several metabolic process, including bone formation, antioxidant activity, and inflammatory response. It is well documented that the antioxidant capacity and immunity of broilers are enhanced in response to the dietary or in ovo administration of L-AA. The aim of the current study was to determine the effects the in ovo or dietary administration of L-AA on the post hatch performance, plasma nitric oxide, and eye L-AA concentrations of Ross 708 broilers. At 17 days of incubation, the following treatments were administered: non-injected or sham-injected (100 μL of saline) controls, or the injection of either 12 or 25 mg of L-AA suspended in 100 μL of saline. In addition, chicks were fed a commercial diet with or without 200 mg/kg of supplemental L-AA. The results of the current study revealed that the eye concentrations of L-AA were higher in males in comparison to females. Additionally, 12 mg of L-AA decreased plasma nitric oxide levels in 14-day-old male broilers in comparison to all the other in ovo injection treatment groups. Compared to a commercial diet, dietary L-AA lowered feed conversion ratio during the second week of post hatch growth. In conclusion, the in ovo administration of L-AA was more effective in reducing inflammatory response, whereas the dietary source had a greater impact on the live performance of broilers. Further research is needed to determine the aforementioned effects throughout the entire growing phase.

**Abstract:**

Effects of the dietary and in ovo administration of L-ascorbic acid (L-AA) on the performance, plasma nitric oxide, and eye L-AA concentrations of Ross 708 broilers were investigated. At 17 days of incubation, live embryonated hatching eggs were randomly assigned to a non-injected or sham-injected (100 μL of saline) control group, or a group injected with either 12 or 25 mg of L-AA suspended in 100 μL of saline. Chicks received a commercial diet with or without 200 mg/kg of supplemental L-AA and were randomly assigned to each of 6 replicate floor pens in each in ovo injection-dietary treatment combination. Weekly live performance variables through 14 days of post hatch age (doa) and the eye weights in both sexes at 0, 7, and 14 doa were determined. At 0 and 14 doa, plasma nitric oxide levels and eye L-AA concentrations of one bird of each sex in each pen were determined. Dietary supplemental L-AA decreased feed intake and growth between 0 and 7 doa, but from 8 to 14 doa; all birds fed supplemental L-AA had a lower feed conversion ratio. At 14 doa, male chicks had higher eye L-AA concentrations and lower plasma nitric oxide levels when treated in ovo with 12 mg of L-AA. In conclusion, dietary L-AA may be used to improve feed conversion in the second week of broiler post hatch growth. However, the in ovo administration of 12 mg of L-AA can increase male eye L-AA concentrations and is effective in reducing their general inflammatory response.

## 1. Introduction

Vitamin C, also known as L-ascorbic acid (**L-AA**), is a potent antioxidant and immunomodulatory agent in poultry [1,2]. A two-step oxidation reaction is required to convert L-AA to the most available form of L-AA, dehydroascorbic acid [3]. The inter-cellular transport of amino acids is facilitated by dehydroascorbic acid, facilitative glucose transporters, and sodium-dependent vitamin C transporter 2 [4,5]. In chickens, L-AA is synthesized in several tissues including the kidney through the glucuronatexylulose-xylulose cycle in the presence of L-gluconolactone oxidase [4]. Although chickens can biosynthesize L-AA, supplemental L-AA has been shown to improve the performance, antioxidant capacity, and immunity of broilers [6,7]. It is well documented that the inclusion of 200 to 400 mg/kg of L-AA in feed results in an increase in the antioxidant activity of broilers. This coincides with significantly lower levels of malondialdehyde and higher activities of superoxide dismutase and catalase [8]. Furthermore, the supplementation of feed with 200 mg/kg of L-AA has been shown to improve the body weight, body weight gain, and feed intake of broilers and to reduce their mortality when they are subjected to high stocking densities [9]. A study conducted by Peebles and Brake [10] showed that 50 or 100 ppm of dietary L-AA resulted in increased body weight gain and improved nutrient utilization in broiler breeders. Furthermore, an increase in plasma L-AA concentrations has been reported to occur in response to 250 to 2000 μg/mL of supplemental L-AA in water [11]. At low ambient temperatures, 250 or 500 ppm of dietary L-AA has been shown to improve feed efficiency, feed consumption, and hen-day egg production in Japanese Quail [12]. Additionally, supplemental L-AA at 250 mg/kg concentrations in feed has been found to improve the feed intake, body weight gain, feed efficiency, antioxidant activity, and performance of broilers during heat stress [13].

In ovo injection has been widely used in the US poultry industry for the vaccination of broilers against Marek’s disease [14]. Maximum vaccine efficacy is achieved when amniotic injections are administrated between 17.5 and 19.25 days of incubation (**doi**) [14,15,16]. In ovo injection has been shown to be less labor intensive, and it is relatively less stressful for the embryo compared to field vaccination [14,15]. Furthermore, in ovo injection has been used to successfully deliver different nutrients such as vitamins, minerals, carbohydrates, and amino acids to broiler embryos [16]. In ovo injection of vitamins has shown promising results for broiler hatchability [16,17,18] post hatch performance [2,16,17,19], breast meat yield [19,20], meat quality [2,16], immunity [16,19], and antioxidant activity [2,16]. Zhang et al. [16] and Mousstaaid et al. [21] reported that the in ovo injection of 100 μL of L-AA in a dosage range from 0.5 to 25 mg/mL was safe and resulted in no negative effects on hatchability and hatchling quality. Additionally, a decrease in non-enzymatic antioxidant activity was observed in response to 36 mg/mL of L-AA [2], and the in ovo administration of 12 mg/mL of L-AA resulted in a greater feed efficiency in broilers in comparison to those that received saline injections or 36 mg/mL of supplemental L-AA.

An appreciable level of L-AA (1.33 mg/g wet weight) has been found in the corneal epithelium of humans [22]. L-ascorbic acid is a well-known anti-inflammatory as well as antioxidant agent and it is well documented that L-AA can reduce corneal inflammation in humans [23,24]. Ocular inflammation has been reported in 28 day old broilers exposed to 25 to 70 ppm of aerial ammonia [25], and an increase in atmospheric ammonia levels was also shown to be associated with conjunctivitis (inflammation of conjunctivae) and damage to the cornea of eyes due to an increase in oxidative stress and pro-inflammatory reactions [25,26,27]. However, the concentration of L-AA in the eyes of chickens and its relationship to inflammation have not been previously reported. Therefore, the objectives of the current study were to determine the effects of the dietary and in ovo administration of L-AA on the performance, plasma nitric oxide (**NO**), and eye L-AA concentrations of Ross 708 broilers at 0 and 14 days of post hatch age (**doa**).

## 2. Material and Method

### 2.1. Egg Incubation

This experimental protocol was approved by the Institutional Animal Care and Use Committee of Mississippi State University (Protocol #IACUC-20-248). Sixty Ross 708 broiler hatching eggs were set in a single tray replicate flat belonging to each of the four treatment groups that were randomly arranged on each of the six replicate tray levels (1440 total eggs) in a single-stage incubator (Chick Master Incubator Company, Medina, OH, USA). To prevent the positional effects of the treatment groups in the incubator, the treatment groups on each tray level were randomly rearranged. The same incubator served as both a setter and hatcher unit. The incubation conditions were applied in accordance with the procedures described by Fatemi et al. [28]. All experimental eggs were candled at 11 and 17 doi according to the procedure of Ernst et al. [29]. At 17 doi, 100 μL solution volumes of each pre-specified treatment were freshly prepared immediately prior to injection in accordance with the procedures described by Mousstaaid et al. [21], and solutions were injected into eggs using a Zoetis Inovoject m (Zoetis Animal Health, Research Triangle Park, NC, USA) multi-egg injection machine. In ovo injection treatment were: (1) non-injected control; (2) injection of 50 μL of saline (sham-injection control); or injection of 100 μL of saline containing (3) 12 mg of L-AA (**L-AA 12**), or (4) 25 mg of L-AA (**L-AA 25**). The form and source of L-AA used in this study were the same as those used by Zhang et al. [17] and Mousstaaid et al. [21]. All in ovo injection solutions were prepared according to the procedures described by Zhang et al. [17]. After injection, eggs were transferred to the hatchling baskets according to the method described by Fatemi et al. [18,20].

### 2.2. Hatchability and Hatch Residue

At hatch (21 doi), the group weight of all chicks belonging to a replicate basket representative of each treatment-replicate group was determined and the birds were counted to determine mean hatchling body weight (**BW**). Furthermore, the hatchability of injected live embryonated eggs (**HI**) was determined according to the procedure described by Peebles et al. [30]. Hatch residue was also analyzed as described by Ernst et al. [29]. Late, dead-pip, live-pip and hatchling mortalities were defined, respectively, as those that occurred between 17 and 21 doi prior to pip, an external full pip but dead in the shell, an external full pip that was live in the shell but failed to complete full shell emergence, and dead immediately after complete emergence from the shell [18,19,28]. Hatch residue analysis revealed that late, pip, and post-pip mortalities did not differ significantly (*p* > 0.05) among treatments.

### 2.3. Post Hatch Grow Out

For the post hatch period, chicks from each of the six replicate tray levels that belonged to the same treatment within a hatchling basket were combined into a common pool. Upon feather sexing, 9 male and 9 female chicks were randomly selected from the pool and were placed in each of the six replicate floor pens in each of the in ovo injection-dietary treatment group combinations (4 in ovo injections × 2 dietary treatments × 6 replicate pens = 48 total pens). There was a 0.062 m^2^/bird stocking density in each floor pen that contained litter previously used by broiler flocks and was top-dressed with fresh wood shavings. All birds received a Mississippi State University basal corn-soybean diet formulated to meet Ross 708 commercial guidelines [20,28], or the same diet supplemented with 200 mg/kg of L-AA (Table 1). The calculated and analyzed (HPLC employed by an independent lab) L-AA concentrations used in the standard commercial and L-AA-supplemented commercial diets are provided in Table 2. Feed and water were provided for ad libitum consumption throughout the 14 doa period.

All birds were fed a starter diet from 0 to 14 doa, and weekly performance variables including, BW, BW gain (**BWG**), average daily BW gain (**ADG**), feed intake (**FI**), average daily FI (**ADFI**), feed conversion ratio (**FCR**; g feed/g gain) and mortality were determined between 0 and 7, 8 and 14, and 0 and 14 doa. There were only three bird mortalities observed in the 0 to 7 doa period. Therefore, FCR in that age interval was adjusted for bird mortality. The performance data were based on approximately 18 birds per treatment-replicate pen for the 0 to 7 doa period and on approximately 16 birds per treatment-replicate pen for the 8 to 14 and 0 to 14 doa periods.

### 2.4. Eye Weight and Plasma and Eye L-Ascorbic Acid Concentrations

At 0, 7, and 14 doa, 12, 6, and 6 birds, respectively, per treatment (one male and one female bird per treatment-replicate pen) were randomly selected to be individually weighed and bled according to the method described by Mousstaaid et al. [21] and Fatemi et al. [31,32], and then birds were euthanized via carbon dioxide. Plasma was subsequently extracted according to the procedure described by Fatemi et al. [31,32]. The BW and the combined absolute weights of both whole eye samples (**EYW**) from one male and one female bird in each replicate pen were determined. Relative eye weights ((**REYW)**; (eye wt/BW) × 100) were also reported as a percentage of BW. Eye L-AA concentrations were determined for samples collected at 0 and 14 doa. Both eyes at 0 doa, and the right eye at 14 doa, of each bird were gently washed with 0.01 M PBS and was then homogenized. A protein extraction reagent (MyBioSource, San Diego, CA, USA) was added during homogenization using a gentle homogenizer (Cole-Parmer, Vernon Hills, IL, USA). Samples were kept cold on ice during preparation. After blending, samples were centrifuged for 10 min at 1000–3000 rpm and then stored at −20 °C. Plasma and eye L-AA concentrations were assayed using a Chicken Vitamin C (VC) Elisa kit (MyBioSource, San Diego, CA, USA). Optical density at 450 nm (OD450) for L-AA concentration in eye and blood were measured with a SpectraMax M5 Microplate Reader (Molecular Devices, San Jose, CA, USA).

### 2.5. Inflammatory Response

At 0 and 14 doa, inflammatory analysis was performed for birds in each of the 6 replicate pens within each of the 8 treatment combinations (4 in ovo × 2 dietary). Approximately 0.2 to 0.5 mL of plasma was used for immunological assay. Plasma nitric oxide (**NO**) concentration was determined according to the manufacturer’s protocol (Cayman Chemical Company, Ann Arbor, MI, USA), and followed the procedure described by Bowen et al. [33] and Fatemi et al. [32]. The optical density (450 nm) for each NO sample was measured with a SpectraMax M5 Microplate Reader (Molecular Devices, San Jose, CA, USA). The concentration of NO was subsequently calculated according to the following formula provided by the manufacturer (Cayman Chemical Company, Ann Arbor, MI, USA):$$\mathrm{Nitric}\;\mathrm{oxide}\;\left(\mu M\right)=\left[\frac{A450-\left(y-\mathrm{intercept}\right)}{\mathrm{slope}}\right]\times \frac{200\;\mu L}{\mathrm{volume}\;\mathrm{of}\;\mathrm{sample}\;\mathrm{used}\;\left(80\;\mu L\right)}\times \mathrm{dilution}$$

### 2.6. Statistical Analyses

The experimental unit was the incubator hatch basket for the hatch data and the floor pen for the performance, individual body and eye weights, and eye L-AA and plasma NO concentrations data. The experimental design was a randomized complete block for both the incubational and rearing periods, and all data were analyzed separately within each time period. There were four in ovo injection treatments for the incubation period, and eight treatment combinations (4 in ovo × 2 dietary) in the grow-out period. In the incubation phase, tray level was the blocking factor, with all in ovo injection treatments randomly represented on each of the 8 levels. In the grow-out phase, a group of pens was the blocking factor, with each dietary-in ovo injection treatment combination being randomly and equally represented in each of the six replicate blocks of pens. A 2 dietary × 4 in ovo factorial arrangement of treatments was used to analyze the post hatch performance data. To test for the main or interactive treatment effects for the hatch and post hatch performance data, the procedure for linear mixed models (PROC MIXED) of SAS©, version 9.4 (SAS Institute Inc., Cary, NC, USA) [34,35,36,37] was used. The performance data were analyzed according to the following model:Y_ij_ = μ + B_i_+ T_j_ + E_ij_
where μ was the population mean; Bi was the blocking factor (i = 1 or 2); T_j_ was the effect of each in ovo injection treatment (j = 1 to 4); and Eij was the residual error.

A 4 in ovo × 2 dietary × 2 sex factorial analysis in a split-plot experimental design with in ovo and dietary treatments as the whole plots and sex as the sub-plot was used to analyze plasma and eye L-AA concentration and NO concentration data at 14 doa. To test for the main and interactive treatment effects for the above variables, the procedure for linear mixed models (PROC GLIMMIX) of SAS 9.4© [34,35,36,37] was used. The data were analyzed according to the following model:Y_ijk_ = μ + B_i_+ D_j_ + I_k_ + (DI)_jk_ + S_l_ + (DS_il_)+ (IS_kl_) + (DIS_ilk_)+ E_ijkl_
where μ was the population mean; B_i_ was the block factor (i = 1 to 2); D_i_ was the effect of each dietary treatment (j = 1 to 2); I_k_ was the effect of in ovo injection treatment (k = 1 to 4); (DI)_ij_ was the interaction of each dietary treatment with in ovo injection treatment; S_l_ was the effect of each sex treatments (l = 1 to 2), which was the sub plot factor; (DS)_il_ was the interaction of each dietary treatment with sex treatment; (IS)_kl_ was the interaction of each in ovo injection treatment with sex treatment; (DS)_il_ was the interaction of each dietary treatment with in ovo injection treatment and sex treatment; and E_ijkl_ was the residual error. All data were analyzed using ANOVA. The results for all data are shown as mean ± SEM and means separations were performed by Fisher’s protected least significant difference [38]. Differences in means were considered statistically significant at *p* ≤ 0.05.

## 3. Results

### 3.1. Hatch and Broiler Performance

Interaction means were provided only in those tables in which significant treatment interactions were noted. There were no significant differences between the in ovo injection treatments for HI, hatch (late, dead-pip, live-pip, and hatchling) residue, and hatchling BW data (Table 3). Percent mortality between 0 and 7, 8 and 14, and 0 and 14 doa was not affected by in ovo or dietary treatment, and there were also no significant main or interactive effects due to in ovo or dietary treatment for 14 doa BW (Table 4). However, there was a significant main effect due to dietary treatment for 7 doa BW, in which birds fed the L-AA diet had a lower BW in comparison to those fed the standard commercial diet (Table 4). Likewise, between 0 and 7 doa, there were significant main effects due to diet for BWG, ADG, FI, and ADFI, and there was a significant main effect due to in ovo treatment for FI, ADFI, and FCR (Table 4). Birds fed the L-AA diet had a lower BWG, ADG, FI, and ADFI. In addition, between 0 and 7 doa, the L-AA 12 in ovo treatment resulted in higher FI, ADFI, and FCR values in comparison to those in the non-injected and L-AA 25 in ovo treatment groups. Furthermore, FI and ADFI in the saline injection treatment were higher than those in the non-injected treatment, and the FCR in the saline injection treatment was higher than that in the L-AA 25 treatment.

Conversely, between 8 and 14 doa, there were only significant main effects due to diet and in ovo treatment for FCR (Table 4). The L-AA 25 in ovo treatment resulted in a higher FCR in comparison to the non-injected and saline-injected treatments, with the L-AA 12 treatment intermediate, and the L-AA-supplemented diet resulted in a lower FCR compared to the un-supplemented commercial diet. Between 0 and 14 doa, there were significant main effects due to diet for FI and ADFI, and due to in ovo treatment for FCR (Table 4). Dietary L-AA resulted in a lower FI and ADFI in comparison to the standard commercial diet. Additionally, the in ovo L-AA 12 treatment resulted in a higher FCR in comparison to the non-injected treatment, with the saline and L-AA 25 in ovo injection treatments intermediate. There were no significant interactions observed between diet and in ovo injection treatment for BW at 7 or 14 doa or for any of the broiler performance variables between 0 and 7, 8 and 14, and 0 and 14 doa (Table 4).

### 3.2. Eye Weight and L-Ascorbic Acid Concentration

At 0 doa, there was a significant main effect due to sex for BW, EYW, and REYW, and a significant main effect due to in ovo treatment for BW (Table 5). The BW, EYW, and REYW of males were significantly greater than those of females, and across sex, the BW of birds in the non-injected, saline-injected, and L-AA 12 in ovo treatments were greater than those in the L-AA 25 treatment. At 7 doa, there was a significant main effect due to sex for EYW, and there was a significant main effect due to post hatch diet for BW and REYW (Table 5). Across the in ovo and dietary treatments, the EYW of males was greater than that of females, and across sex and in ovo treatment, the L-AA-supplemented diet resulted in a lower BW and a greater REYW in comparison to those in the commercial diet. At 14 doa, there was a significant main effect due to sex for BW and EYW, a significant main effect due to in ovo treatment for EYW, and a significant main effect due to diet for REYW (Table 5). The BW and EYW values of males were significantly higher in comparison to those of females, and those birds across sex and dietary treatment that were injected with 25 mg of L-AA had a lower EYW in comparison to those in any of the other in ovo-injected treatment groups. Furthermore, birds across sex and in ovo treatment fed commercial diets had a higher REYW in comparison to those fed L-AA-supplemented diets. There were no significant interactive effects among sex, in ovo treatment, or dietary treatment for BW, EYW, or REYW at 0, 7, and 14 doa.

At hatch (0 doa), significant differences in eye L-AA concentration were observed between male and female hatchlings (Table 6). Male hatchlings had a higher eye L-AA concentration when compared to female hatchlings. However, there was no significant difference among in ovo injection treatments and there was no significant interaction between sex and in ovo injection treatment for eye L-AA concentration at hatch. At 14 doa, there was a significant interaction between sex and in ovo injection treatment for eye L-AA concentration (Table 7). Eye L-AA concentration was significantly higher in males belonging to the L-AA 12 treatment compared to that in non-injected and saline-injected males and in females, regardless of in ovo treatment. Furthermore, eye L-AA concentrations in males belonging to the L-AA 25 in ovo treatment were higher than those in females that belonged to the saline-injected and L-AA 12 in ovo treatments. However, no significant interaction was observed between diet, sex, and in ovo injection treatment for eye L-AA concentration at 14 doa (Table 7).

### 3.3. Inflammatory Response

No significant main effect due to sex or in ovo treatment, and no significant interaction between sex and in ovo injection treatment was observed for plasma NO concentration at 0 doa (Table 6). However, there was a significant interaction between sex and in ovo injection treatment for plasma NO level at 14 doa (Table 7). In males, those that were non-injected or were in ovo-injected with saline or 25 mg of L-AA had higher plasma NO levels compared to males that belonged to the L-AA 12 treatment group and to females belonging to all the in ovo injection treatment groups. Additionally, at 14 doa, no significant interaction for plasma NO concentration was observed between sex and diet, in ovo and dietary treatment, or between diet, sex, and in ovo injection treatment (Table 7).

## 4. Discussion

The aim of this study was to determine the effect of in ovo and dietary supplementation of L-AA on early straight run broiler performance, and the post hatch inflammatory response, and eye L-AA concentrations in both male and female broilers. Dietary L-AA reduced broiler growth (BW and ADG) between 0 and 7 doa because of concomitant reductions in FI and ADFI, but supplemental dietary L-AA reduced or improved FCR between 8 and 14 doa. Nevertheless, although there were overall decreases in FI and ADFI between 0 and 14 doa, no overall concomitant effect on growth was observed in that age interval. Conversely, Lohakare et al. [7] reported in a previous study that dietary L-AA at 200 mg/kg increased the FI and improved the BWG and FCR of commercial broilers between 0 and 21 doa. Dietary supplementation of 200 mg/kg of L-AA and 70 mg/mL of zinc has been shown to increase the BW and reduce the FCR of Arbor Acres broilers [39], and a diet supplemented with 200 mg of L-AA has been shown to improve the early post hatch growth rate and feed utilization in broilers subjected to heat stress [40]. Therefore, in association with more efficient feed usage in the birds of those studies in response to supplemental dietary L-AA, their growth was observed to also increase. According to the literature, supplemental L-AA has not displayed any positive effects on the live performance of broilers during the first week of post hatch, while the majority of improvements were observed in the grower or finisher phase [7,39,40]. The lack of efficacy of dietary L-AA could be linked to the immaturity of the digestive system, which will be fully developed after 10 to 12 days of post hatch age [41].

Improvements in broiler performance in response to dietary L-AA may be due to the enhancement of adaptive immunity, small intestine morphology, and energy digestibility. The number of CD4 lymphocytes has been shown to increase in birds that were fed 100 to 200 mg/kg of dietary L-AA in comparison to commercial diet-fed broilers [7]. The CD4 cells are linked to T-helper and regulatory T cells [42]. The mechanism of action of these cells is to inhibit T cell proliferation and cytokine production in order to suppress the immune response, thereby maintaining immune homeostasis [43,44]. In addition to immunity, regardless of the level of inclusion, dietary L-AA has also been shown to improve small intestine morphology [45,46]. Supplemental dietary L-AA has resulted in a higher epithelial villus height as well as a greater villus surface area in the duodenum of Ross 308 broilers. A longer villus height is associated with an increased intestinal absorptive surface area, leading to increased nutrient absorption [31,32,47], an increase in BWG [48], and a decreased FCR [49]. It is well documented that energy digestibility has also increased in response to the use of 200 mg of dietary L-AA [7]. Therefore, improvements in the overall live performance of broilers fed 200 mg/kg of L-AA in the other earlier studies may be linked to an increase in energy digestibility and improvements in their small intestine morphology and cell-mediated immunity. Differences in the growth response of the broilers in the current study in contrast to the other earlier studies may be related to differences in broiler strain, basal diet, post hatch period, or environmental conditions.

In the current study, the in ovo injection of different levels of L-AA resulted in an inconsistent pattern of live performance within the first two weeks of post hatch growth. During the first week post hatch, the in ovo injection of 25 mg of L-AA showed promising results concerning its effects on broiler live performance; however, this advantage disappeared in the second week. This inconsistency could be because the live performance of the broilers was only examined for two weeks rather than over a longer period. The lowest level of in ovo injected L-AA used in the current study was 12 mg. However, it is well documented that there are positive effects of the in ovo injection of lower dosages of L-AA on the live performance of broilers after 14 doa [2,50,51]. The in ovo administration of L-AA between 1 and 6 mg has been earlier shown to improve broiler performance [2,50,51]. As compared to a saline-injected control group, the in ovo injection of L-AA between 3 and 6 mg has also been shown to increase the BW and BWG of broilers between 14 and 21 doa [2]. Zhu et al. [51] similarly observed that broilers that received an in ovo injection of 1 mg of L-AA experienced a higher BWG in comparison to a saline-injected control group. Furthermore, BW, ADG, and FCR were improved in response to the in ovo injection of 3 mg of L-AA compared to the non-injected control group in that study. The contrast in performance results between this study and the earlier studies in response to in ovo L-AA administration may be partially related to the shorter post hatch period examined in the current study.

Eye L-AA concentrations were currently higher in male compared to female broilers at hatch and at 14 doa. Additionally, eye L-AA concentrations were highest in males that were injected with 12 mg of L-AA compared to all other in ovo injection controls in both sexes. These eye concentration results were not directly associated with those of eye weight, as only the L-AA 25 in ovo treatment across sex resulted in a reduction in EYW at 14 doa. Furthermore, only supplemental L-AA in the diet resulted in significant changes in REYW, in that across sex and in ovo treatment, dietary supplemental L-AA increased REYW at 7 doa, whereas it decreased REYW at 14 doa. To date, there has been no research conducted to measure and compare L-AA concentrations in the eyes and other organs of male and female chickens. However, when compared to males, the concentration of L-AA has been found to be lower in the plasma and to be higher in the urine and spleens of female mice [52]. In addition, the in ovo injection of 12 mg of L-AA resulted in the highest eye L-AA concentrations in the current study. The production of L-AA has been shown to be limited by decreased levels of L-gulonolactone oxidase activity [53]. Therefore, Kuo et al. [52] concluded that the noted differences could be due to differences in the level of expression of gulonolactone oxidase in various tissues of the organism. The higher concentration of L-AA in male broilers in this study may be subsequently due to higher levels of L-gulonolactone oxidase activity in their eyes. Further research is required to determine the effects of the in ovo administration of different levels of L-AA on L-gulonolactone oxidase levels in various organs of male and female broilers.

The production of NO has been shown to increase during a systemic inflammation reaction by the oxidation of L-arginine and the action of NO synthase [54]. Plasma NO levels were decreased because of the in ovo injection of 200 mg of L-AA in this study. Therefore, the in ovo injection of 12 mg of L-AA might have the potential to reduce systemic inflammation in broiler chickens. Furthermore, an increase in ocular oxidative stress has been shown to be associated with an increase in ocular lesion and inflammation [26,27]. It is well documented that stress levels increase when broilers have experienced high aerial ammonia levels or when subjected to excessive heat, which is often detrimental to poultry health and performance [55,56,57]. In ovo supplementation of L-AA at 12 or 36 mg has been shown to enhance enzymatic (superoxide dismutase) and non-enzymatic (malondialdehyde) antioxidant activities, with concomitant improvements in broiler performance [2]. Supplemental L-AA has also been shown to improve feed intake, weight gain and to reduce oxidative stress [58,59]. Improvements in bird vision in association with reduced incidences of corneal lesions may allow for increased feed intake and subsequent growth. Therefore, the in ovo injection of higher dosages of L-AA in combination with a subsequent reduction in supplemental dietary L-AA below 200 mg/kg could hypothetically reverse the reduction of feed intake and BW gain between 0 and 7 doa that were observed in response to the 200 mg/kg level of dietary L-AA. It is also worth mentioning that a decrease in ventilation rate has been used to allow for increased heat retention in a broiler facility during a cold season but may result in increased harm to poultry health [60]. It is expected that in ovo injections of L-AA have even greater potential to reduce ocular lesion incidence and to enhance the live performance of broilers when they are subjected to oxidative stress due to high levels of atmospheric ammonia.

## 5. Conclusions

In conclusion, the effects of the in ovo and dietary administration of L-AA on the early growth performance of straight run broilers and the systemic inflammation and eye L-AA concentrations of male and female broilers were investigated. The results of the current study showed that reductions in feed intake in response to supplemental dietary L-AA at a concentration of 200 mg/kg resulted in a poor live performance, but that relationship became opposite in the second week post hatch. Moreover, the in ovo injection of 12 mg of L-AA resulted in higher eye L-AA concentrations and a lower inflammatory response in male broilers at 14 doa. The partial improvements observed for either the dietary or in ovo administration of L-AA could be linked to an enhanced immunity and antioxidant capacity. Moreover, although the in ovo administration of 12 or 25 mg of L-AA was not effective in improving the performance of male or female broilers between 0 and 14 doa, the in ovo injection of 12 mg of L-AA resulted in higher eye L-AA concentrations and a lower inflammatory response in male broilers at 14 doa. Further study is needed to determine effects of higher dosages of in ovo-injected L-AA and longer durations of the provision of supplemental dietary L-AA on the live performance, eye L-AA concentrations, and immunity of broilers under normal and challenging conditions.

## Figures and Tables

**Table 1 animals-12-02573-t001:** Feed composition of the experimental starter diets from 0 to 14 days of post hatch age.

Item	Commercial Diet	L-Ascorbic Acid-Supplemented Diet ^1^
Ingredient	--------------------------------(%)-------------------------------------	
Yellow corn	53.23	53.23
Soybean meal	38.23	38.23
Animal fat	2.60	2.60
Dicalcium phosphate	2.23	2.23
Limestone	1.27	1.27
Salt	0.34	0.34
Choline chloride 60%	1.00	1.00
Lysine	0.28	0.28
DL-Methionine	0.37	0.37
L-threonine	0.15	0.15
Premix ^2^	0.25	0.25
Coccidiostat ^3^	0.05	0.05
Total	100	100
Calculated nutrients		
Crude protein	23	23
Calcium	0.96	0.96
Available phosphorus	0.48	0.48
Apparent metabolizable energy (AME; Kcal/kg)	3000	3000
Digestible Methionine	0.51	0.51
Digestible Lysine	1.28	1.28
Digestible Threonine	0.86	0.86
Digestible total sulfur Amino acids (TSAA)	0.95	0.95
Sodium	0.16	0.16
Choline	0.16	0.16

^1^ Broiler premix supplemented with 200 mg of L-Ascorbic acid (Vitamin C) per kilogram of diet (DSM, Parsippany, NJ, USA). ^2^ The broiler premix provided per kilogram of diet: vitamin A (retinyl acetate), 10,000 IU; cholecalciferol, 4000 IU; vitamin E (DL-α-tocopheryl acetate), 50 IU; vitamin K, 4.0 mg; thiamine mononitrate (B_1_), 4.0 mg; riboflavin (B_2_), 10 mg; pyridoxine HCL (B_6_), 5.0 mg; vitamin B_12_ (cobalamin), 0.02 mg; D-pantothenic acid, 15 mg; folic acid, 0.2 mg; niacin, 65 mg; biotin, 1.65 mg; iodine (ethylene diamine dihydroiodide), 1.65 mg; Mn (MnSO_4_H_2_O), 120 mg; Cu, 20 mg; Zn, 100 mg, Se, 0.3 mg; Fe (FeSO_4_.7H_2_O), 800 mg. ^3^ Decocx ® (Zoetis, Parsippany, NJ, USA).

**Table 2 animals-12-02573-t002:** The calculated and analyzed values of L-ascorbic acid (L-AA) concentrations of standard commercial and L-AA-supplemented commercial diets fed in the starter dietary phase.

Diet	L-AA Calculated	L-AA Analyzed
	---------------------------------------mg/kg-------------------------------------	
Commercial	0	ND ^1^
L-AA supplemented ^2^	200	150

^1^ Not detected; the concentration of nutrients was less than 2 μg, which was not at the detectable level. ^2^ Diet supplemented with L-AA throughout the rearing period. *n* = Samples were analyzed in triplicate for the means calculations of the analyzed levels of supplemented L-AA.

**Table 3 animals-12-02573-t003:** Effects of treatment non-injected; saline-injected (saline); saline containing 12 mg of L-ascorbic acid (L-AA 12), or 25 mg of L-ascorbic acid (L-AA 25) across dietary treatment on hatchability of injected live embryonated eggs (HI), hatchling body weight (BW), and hatch residue analysis variables at 21 days of incubation (doi).

Treatment	HI	Late ^1^	Dead-Pip ^2^	Live-Pip ^3^	Hatchling ^4^	Hatchling BW (g)
	--------------------------------%----------------------------------					
Non-injected ^5^	91.6	1.53	1.18	0.05	0.32	42.0
Saline ^6^	90.0	3.17	1.18	0.07	0.00	42.3
L-AA 12 ^7^	91.8	3.25	0.57	0.03	0.30	41.4
L-AA 25 ^8^	94.9	2.58	0.27	0.02	0.00	42.0
SEM	2.53	0.871	0.658	0.029	0.308	0.81
*p*-value	0.305	0.210	0.425	0.368	0.582	0.756

^1^ Mortality between 17 and 21 doi, prior to pip. ^2^ Mortality when embryo had an external pip but dead in the shell. ^3^ Mortality when embryo had an external pip that was live in the shell but failed to complete full shell emergence. ^4^ Mortality immediately after complete emergence of hatchlings from the shell. ^5^ Eggs that were not injected. ^6^ Eggs that were injected with 100 μL saline at 17 doi. ^7^ Eggs that were injected with 100 μL saline containing L-AA 12 at 17 doi. ^8^ Eggs that were injected with 100 μL saline containing L-AA 25 at 17 doi. *n* = Approximately 60 eggs in each of 6 tray replicate groups in each treatment were used for means calculations.

**Table 4 animals-12-02573-t004:** Effects of treatment non-injected; saline-injected (saline); saline containing 12 mg of L-ascorbic acid (L-AA 12), or 25 mg of L-ascorbic acid (L-AA 25) at 17 days of incubation (doi) and either a commercial diet or a diet supplemented with 200 mg/kg of L-AA on live performance variables.

Treatment		7 doa	---------------------------0 to 7 doa ---------------------------					
		BW ^1^ (g)	BWG ^1^ (g)	ADG ^1^ (g)	FI ^1^ (g)	ADFI ^1^ (g)	FCR ^1^ (g/g)	Mortality(%)
In ovo								
	Non-injected ^2^	117	73.4	10.5	174 ^c^	24.8 ^c^	2.40 ^bc^	0.56
	Saline ^3^	117	74.2	10.6	188 ^ab^	26.9 ^ab^	2.56 ^ab^	0.56
	L-AA 12 ^4^	117	73.7	10.5	191 ^a^	27.3 ^a^	2.69 ^a^	0
	L-AA 25 ^5^	121	79.1	11.3	179 ^bc^	25.5 ^bc^	2.28 ^c^	0.56
	SEM	3.2	3.19	0.45	4.2	0.84	0.135	1.067
Diet								
	Commercial	122 ^a^	79.7 ^a^	11.4 ^a^	190 ^a^	27.2 ^a^	2.44	0.69
	L-AA ^6^	114 ^b^	70.5 ^b^	10.1 ^b^	176 ^b^	25.1 ^b^	2.52	0
	SEM	2.3	2.25	0.32	2.9	0.59	0.095	0.755
*p*-value								
	In ovo	0.556	0.254	0.277	0.016	0.017	0.026	0.486
	Diet	0.001	0.001	0.001	0.001	0.001	0.414	0.382
	In ovo × Diet	0.538	0.537	0.537	0.853	0.851	0.709	0.512
		14 doa	--------------------------8 to 14 doa ---------------------------					
In ovo								
	Non-injected	319	202	28.8	201	28.7	1.21 ^b^	0
	Saline	312	195	27.8	192	27.8	1.22 ^b^	0
	L-AA 12	308	191	27.3	199	28.4	1.26 ^ab^	0
	L-AA 25	319	199	28.4	211	30.1	1.30 ^a^	0
	SEM	11.7	9.5	1.36	9.0	1.29	0.035	-
Diet								
	Commercial	319	197	28.1	204	29.2	1.27 ^a^	0
	L-AA	311	197	28.1	197	28.2	1.22 ^b^	0
	SEM	8.3	6.7	0.96	6.4	0.91	0.025	-
*p*-value								
	In ovo	0.757	0.716	0.717	0.258	0.267	0.047	-
	Diet	0.321	0.974	0.966	0.279	0.278	0.018	-
	In ovo × Diet	0.538	0.606	0.608	0.349	0.347	0.178	-
			-------------------------------0 to 14 doa -----------------------------------					
In ovo								
	Non-injected		275	19.7	442	30.2	1.57 ^b^	0.67
	Saline		269	19.2	429	30.6	1.63 ^ab^	0.67
	L-AA 12		265	18.9	439	31.4	1.71 ^a^	0
	L-AA 25		278	19.8	440	31.4	1.62 ^ab^	0.67
	SEM		11.7	0.84	11.4	0.81	0.047	1.067
Diet								
	Commercial		276	19.7	446 ^a^	31.9 ^a^	1.67	0.83
	L-AA		276	19.1	419 ^b^	29.9 ^b^	1.60	0
	SEM		8.3	0.59	8.1	0.58	0.033	0.755
*p*-value								
	In ovo		0.700	0.693	0.365	0.361	0.040	0.486
	Diet		0.286	0.284	0.002	0.007	0.075	0.382
	In ovo × Diet		0.585	0.574	0.336	0.330	0.645	0.512

^a–c^ Treatment means within the same variable column within the type of treatment with no common superscript differ significantly (*p* < 0.05). ^1^ body weight (BW), the BW gain (BWG), average daily BW gain (ADG), feed intake (FI), average daily FI (ADFI), feed conversion ratio (FCR). ^2^ Eggs that were not injected. ^3^ Eggs that were injected with 100 μL saline at 17 doi. ^4^ Eggs that were injected with 100 μL saline containing L-AA 12 at 17 doi. ^5^ Eggs that were injected with 100 μL saline containing L-AA 25 at 17 doi. ^6^ Commercial diet supplemented with 200 mg/kg L-AA throughout the rearing period. *n* = 18 birds in each of 6 replicate groups in each treatment combination were used for means calculations.

**Table 5 animals-12-02573-t005:** Effects of treatment non-injected; saline-injected (saline); saline containing 12 mg of L-ascorbic acid (L-AA 12), or 25 mg of L-ascorbic acid (L-AA 25) at 17 days of incubation (doi) and either a commercial diet or a diet supplemented with 200 mg/kg of L-AA on the body weight (BW) and eye characteristics of male and female broilers at 0 days of posthatch age.

Treatment		BW ^1^ (g)	EYW ^1^ (g)	REYW ^1^ (%)
	-------------------------------0 doa-----------------------------------			
In ovo				
	Non-injected ^2^	43.3 ^a^	0.574	1.33
	Saline ^3^	42.9 ^a^	0.554	1.30
	L-AA 12 ^4^	42.3 ^a^	0.539	1.27
	L-AA 25 ^5^	40.7 ^b^	0.560	1.38
	SEM	0.65	0.0170	0.048
Sex				
	Male	42.8 ^a^	0.586 ^a^	1.37 ^a^
	Female	41.6 ^b^	0.528 ^b^	1.27 ^b^
	SEM	0.32	0.0120	0.034
*p*-Value				
	In ovo	0.002	0.240	0.147
	Sex	0.031	0.0001	0.005
	In ovo × Sex	0.627	0.947	0.999
		-------------------------------7 doa-----------------------------------		
		BW (g)	EYW (g)	REYW (%)
In ovo				
	Non-injected	129	1.31	1.028
	Saline	126	1.25	1.026
	L-AA 12	119	1.28	1.100
	L-AA 25	133	1.25	0.969
	SEM	4.48	0.028	0.035
Diet				
	Commercial	132 ^a^	1.25	0.969 ^b^
	L-AA ^5^	122 ^b^	1.30	1.092 ^a^
	SEM	3.17	0.020	0.025
Sex				
	Male	130	1.32 ^a^	1.047
	Female	124	1.23 ^b^	1.015
	SEM	3.17	0.018	0.025
*p*-Value				
	In ovo	0.159	0.414	0.082
	Diet	0.025	0.100	0.001
	In ovo × Diet	0.339	0.665	0.414
	Sex	0.199	0.001	0.369
	In ovo × Sex	0.217	0.211	0.098
	Diet × Sex	0.989	0.559	0.860
	In ovo × Diet × Sex	0.923	0.626	0.867
		-------------------------------14 doa-----------------------------------		
Treatment		BW (g)	EYW (g)	REYW (%)
In ovo				
	Non-injected	329	1.80 ^a^	0.542
	Saline	320	1.78 ^a^	0.566
	L-AA 12	304	1.82 ^a^	0.617
	L-AA 25	309	1.63 ^b^	0.512
	SEM	22.7	0.070	0.039
Diet				
	Commercial	318	1.76	0.609 ^a^
	L-AA ^5^	313	1.75	0.549 ^b^
	SEM	10.1	0.036	0.028
Sex				
	Male	335 ^a^	1.82 ^a^	0.560
	Female	296 ^b^	1.68 ^b^	0.599
	SEM	14.2	0.035	0.027
*p*-Value				
	In ovo	0.618	0.050	0.275
	Diet	0.726	0.880	0.038
	In ovo × Diet	0.697	0.568	0.194
	Sex	0.008	0.005	0.151
	In ovo × Sex	0.600	0.535	0.600
	Diet × Sex	0.564	0.144	0.123
	In ovo × Diet × Sex	0.671	0.788	0.967

^a,b^ Treatment means within the same variable column within the type of treatment with no common superscript differ significantly (*p* < 0.05). ^1^ Combined absolute (EYW) and relative (REYW) weights of both eyes. ^2^ Eggs that were not injected. ^3^ Eggs that were injected with 100 μL saline at 17 doi. ^4^ Eggs that were injected with 100 μL saline containing L-AA 12 at 17 doi. ^5^ Eggs that were injected with 100 μL saline containing L-AA 25 at 17 doi. *n* = one male and one female in each of 6 replicate groups in each treatment were used for means calculations.

**Table 6 animals-12-02573-t006:** Effects of treatment non-injected; saline-injected (saline); saline containing 12 mg of L-ascorbic acid (L-AA 12), or 25 mg of L-ascorbic acid (L-AA 25) at 17 days of incubation (doi) on L-AA concentrations in both eyes (Eye L-AA) and nitric oxide (NO) concentrations in the plasma of male and female broilers at 0 days of post hatch age.

		Eye L-AA ^5^	NO ^5^
Treatment		---------------------------------------(μM)---------------------------------------	
In ovo			
	Non-injected ^1^	2.80	6.55
	Saline ^2^	2.69	5.28
	L-AA 12 ^3^	2.64	5.67
	L-AA 25 ^4^	2.74	5.06
	SEM	0.218	0.856
Sex			
	Male	3.47 ^a^	6.17
	Female	1.96 ^b^	5.10
	SEM	0.112	0.428
*p*-value			
	In ovo	0.736	0.329
	Sex	<0.0001	0.085
	In ovo × Sex	0.609	0.764

^a,b^ Treatment means within the same variable column within the type of treatment with no common superscript differ significantly (*p* < 0.05). ^1^ Eggs that were not injected. ^2^ Eggs that were injected with 100 μL saline at 17 doi. ^3^ Eggs that were injected with 100 μL saline containing L-AA 12 at 17 doi. ^4^ Eggs that were injected with 100 μL saline containing L-AA 25 at 17 doi. ^5^
*n* = One male and one female in each of 6 replicate groups in each treatment were used for means calculations.

**Table 7 animals-12-02573-t007:** Effects of treatment non-injected; saline-injected (saline); saline containing 12 mg of L-ascorbic acid (L-AA 12), or 25 mg of L-ascorbic acid (L-AA 25) at 17 days of incubation (doi) and either a commercial diet or a diet supplemented with 200 mg/kg of L-AA on L-AA concentrations in the right eye (Eye L-AA) and nitric oxide (NO) concentrations in the plasma of male and female broilers at 14 days of post hatch age.

Treatment		Eye L-AA	NO
		----------------------------------------(μM)----------------------------------------	
In ovo			
	Non-injected ^1^	3.22	5.36
	Saline ^2^	3.10	6.13
	L-AA 12 ^3^	3.41	4.57
	L-AA 25 ^4^	3.29	5.88
	Pooled SEM	0.183	0.859
Diet			
	Commercial	3.26	5.13
	L-AA ^5^	3.25	5.84
	Pooled SEM	0.130	0.607
Sex			
	Male	3.41 ^a^	7.41 ^a^
	Female	3.11 ^b^	3.56 ^b^
	Pooled SEM	0.140	0.607
Male			
	Non-injected ^1^	3.29 ^bc^	7.63 ^a^
	Saline ^2^	3.27 ^bc^	9.29 ^a^
	L-AA 12 ^3^	3.85 ^a^	4.68 ^b^
	L-AA 25 ^4^	3.54 ^ab^	8.04 ^a^
	Pooled SEM	0.189	0.859
Female			
	Non-injected	3.15 ^bc^	3.09 ^b^
	Saline	2.93 ^c^	2.98 ^b^
	L-AA 12	2.96 ^c^	4.45 ^b^
	L-AA 25	3.04 ^bc^	3.72 ^b^
	Pooled SEM	0.189	0.859
*p*-Value			
	In ovo	0.425	0.280
	Diet	0.968	0.242
	In ovo × Diet	0.974	0.401
	Sex	0.037	<0.0001
	In ovo × Sex	0.011	0.006
	Diet × Sex	0.127	0.185
	In ovo × Diet × Sex	0.352	0.737

^a–c^ Treatment means within the same variable column within the type of treatment with no common superscript differ significantly (*p* < 0.05). ^1^ Eggs that were not injected. ^2^ Eggs that were injected with 100 μL saline at 17 doi. ^3^ Eggs that were injected with 100 μL saline containing L-AA 12 at 17 doi. ^4^ Eggs that were injected with 100 μL saline containing L-AA 25 at 17 doi. ^5^ Commercial diet supplemented with 200 mg/kg L-AA throughout the rearing period. *n* = one male and one female in each of 6 replicate groups in each treatment were used for means calculations.

## Data Availability

None of the data were deposited in an official repository.

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
