# Peer review of "Effects of the In Ovo and Dietary Supplementation of L-Ascorbic Acid on the Growth Performance, Inflammatory Response, and Eye L-Ascorbic Acid Concentrations in Ross 708 Broiler Chickensâ€"

_animals, 2022, doi:10.3390/ani12192573_

Round 1
Reviewer 1 Report
1. In table 3, why the sum for HI and other three mortality percentage (1-3) is not 100%.
2. The titles of all table are too long and should be shorten.
3. In table 4, how do you understand the results that dietary vitamin C addition seems to develop not good effect on growth performance.
4. you can adjust the header content and combine Table 4 ,5 and 6 into one table. the same for table 7, 8 and 9.
5. only plasma NO concentration is not enough to reveal immune response
6. Did you detect VC concentration in plasma?
Author Response
Reviewer 1:
- In table 3, why the sum for HI and other three mortality percentage (1-3) is not 100%.
Answer: The relevant corrections were applied to Table 3 and corrected values were replaced.
- The titles of all table are too long and should be shorten.
Answer: The relevant corrections were applied to Tables 3 to 10.
- In table 4, how do you understand the results that dietary vitamin C addition seems to develop not good effect on growth performance.
Answer: We agree that the results of dietary vit C is controversial for live performance during the first week post-hatch, in which supplemental vit C reduced BW, BWG, FI and increased FCR; however, the live performance of the Vit C-fed bird was observed in the second week. In the 2nd week post-hatch, dietary Vit C decreased FCR in comparison to the commercial diet. The negative results for the first week could be due to the immaturity of the digestive tract, which takes 10 to 12 days to be fully developed. The relevant information were inserted on Lines 434-439
“According to the literature, supplemental L-AA has not displayed any positive effects on the live performance of broilers during the first week of posthatch, while the majority of improvements were observed in the grower or finisher phase [7, 36, 37]. The lack of efficacy of dietary L-AA could be linked to the immaturity of the digestive system, which will be fully developed after 10 to 12 days of posthatch age [38].”
- you can adjust the header content and combine Table 4 ,5 and 6 into one table. the same for table 7, 8 and 9.
Answer: The relevant corrections were applied to Tables 3-9. Authors prefer to keep the current title for table 10 because the title contains valuable information
- only plasma NO concentration is not enough to reveal immune response
Answer: We agree that nitric oxide (NO) is not the best indicator for an entire immune response. But it is a well-known indicator for a systemic inflammatory response. Thus we have made relevant changes throughout the manuscript.
- Did you detect VC concentration in plasma?
Answer: In the current research we did not measure serum/ plasma L-AA (Vit C) concentrations; however, previous studies have shown the changes in serum L-AA concentrations in response to dietary L-AA. In addition, we observed the changes in serum L-AA concentrations in male hatchling broilers in a previous study conducted by Mousstaaid et al. (2022)
Mousstaaid, A.; Fatemi, S.A.; Elliott, K.E.C.; Alqhtani, A.H.; Peebles, E.D. Effects of the in ovo injection of L-ascorbic acid on broiler hatching performance. Animals (Basel). 2022, 12, 1020. doi: https://doi.org/10.3390/ani12081020.
Reviewer 2 Report
Authors investigated the effects of the dietary and in ovo administration of L-ascorbic acid (L-AA) on the performance, plasma nitric oxide, and eye L-AA concentrations of Ross 708 broilers in the manuscript. The subject of this study is suitable for the “Animal” journal. The authors indicated that dietary L-AA might be used to improve feed conversion in the second week of broiler post-hatch growth, and in ovo administration of 12 mg of L-AA could decrease the overall inflammatory response by increasing the eye L-AA concentration in male birds. This is an exciting study, and I suggest a few corrections to increase the scientific value of the manuscript. After these corrections are addressed by the authors, the manuscript can be accepted.
- The confusing presentation of pre-specified treatment groups in the material and method section should be rewritten more understandably.
- How 12 or 25 mg in ovo L-AA injection was determined should be clearly stated (line 127). Similarly, how was the addition of 200 mg/kg L-AA to the ration determined? should be clearly stated (line 153).
- How eyes are taken from chickens should be detailed, following chemical killing or commercial slaughter? and what tool or device was used for the eye to be taken out? (line 183).
- Table 2 should be removed and should be expressed textually in the material and method section.
- The results have been well presented, and the results have been supported by the discussion, but the conclusion part should be improved.
Author Response
Reviewer 2
Comments and Suggestions for Authors
Authors investigated the effects of the dietary and in ovo administration of L-ascorbic acid (L-AA) on the performance, plasma nitric oxide, and eye L-AA concentrations of Ross 708 broilers in the manuscript. The subject of this study is suitable for the “Animal” journal. The authors indicated that dietary L-AA might be used to improve feed conversion in the second week of broiler post-hatch growth, and in ovo administration of 12 mg of L-AA could decrease the overall inflammatory response by increasing the eye L-AA concentration in male birds. This is an exciting study, and I suggest a few corrections to increase the scientific value of the manuscript. After these corrections are addressed by the authors, the manuscript can be accepted.
- The confusing presentation of pre-specified treatment groups in the material and method section should be rewritten more understandably.
Answer: The relevant changes were applied in the Materials and Methods section in which “in ovo injection treatment” was replaced to “pre-specified treatment “
- How 12 or 25 mg in ovo L-AA injection was determined should be clearly stated (line 127). Similarly, how was the addition of 200 mg/kg L-AA to the ration determined? should be clearly stated (line 153).
Answer: The relevant corrections were applied in Material and method section
- How eyes are taken from chickens should be detailed, following chemical killing or commercial slaughter? and what tool or device was used for the eye to be taken out? (line 183).
Answer: The relevant corrections for the methods of euthanasia and eye sample collection were added to the Material and Method section.
- Table 2 should be removed and should be expressed textually in the material and method section.
Answer: It is common practice to show the results of observed and calculated Levels of the feed additives in tabular form in order to show the reliability of the research and also it is more understandable for the reader. Thus, the authors prefer to keep the table rather than present it in the text
- The results have been well presented, and the results have been supported by the discussion, but the conclusion part should be improved.
Answer: The relevant changes were applied in the Conclusion section
“The results of the current study showed that reductions in feed intake in response to supplemental dietary L-AA at a concentration of 200 mg/kg resulted in a poor live performance, but that relationship became opposite in the second week posthatch. Moreover, the in ovo injection of 12 mg of L-AA resulted in higher eye L-AA concentrations and a lower inflammatory response in male broilers at 14 doa. The partial improvements observed for either the dietary or in ovo administration of L-AA could be linked to an enhanced to immunity and antioxidant capacity.”